# Molecular Detection and Characterization of Rickettsia Species in Ixodid Ticks from Selected Regions of Namibia

**DOI:** 10.3390/microorganisms12050912

**Published:** 2024-04-30

**Authors:** Pricilla Mbiri, Ophelia Chuma Matomola, Walter Muleya, Lusia Mhuulu, Azaria Diegaardt, Bruce Howard Noden, Katendi Changula, Percy Chimwamurombe, Carolina Matos, Sabrina Weiss, Emmanuel Nepolo, Simbarashe Chitanga

**Affiliations:** 1Department of Production Animal Studies, School of Veterinary Medicine, Faculty of Health Sciences and Veterinary Medicine, University of Namibia, Private Bag 13301, Windhoek 10005, Namibia; pmbiri@unam.na; 2Department of Preclinical Studies, School of Veterinary Medicine, Faculty of Health Sciences and Veterinary Medicine, University of Namibia, Private Bag 13301, Windhoek 10005, Namibia; cmatomola@unam.na; 3Department of Preclinical Studies, School of Veterinary Medicine, University of Zambia, P.O. Box 32379, Lusaka 10101, Zambia; muleyawalter@gmail.com; 4Department of Human Biology and Translational Medicine, School of Medicine, Faculty of Health Sciences and Veterinary Medicine, University of Namibia, Private Bag 13301, Windhoek 10005, Namibia; lmhuulu@unam.na (L.M.); adiegaardt@unam.na (A.D.); enepolo@unam.na (E.N.); 5Department of Entomology and Plant Pathology, Oklahoma State University, Stillwater, OK 74078, USA; bruce.noden@okstate.edu; 6Department of Paraclinical Studies, School of Veterinary Medicine, University of Zambia, P.O. Box 32379, Lusaka 10101, Zambia; katendi.changula@gmail.com; 7Department of Natural and Applied Sciences, Namibia University of Science & Technology, Windhoek 10005, Namibia; pchimwamurombe@nust.na; 8Centre for International Health Protection, Robert Koch Institute, 13353 Berlin, Germany; matosc@rki.de (C.M.); weisss@rki.de (S.W.); 9Department of Biomedical Sciences, School of Health Sciences, University of Zambia, P.O. Box 50110, Lusaka 10101, Zambia

**Keywords:** Rickettsiae, *Rickettsia africae*, *Rickettsia aeschlimannii*, Ixodid, ticks, Namibia

## Abstract

Rickettsial pathogens are among the emerging and re-emerging vector-borne zoonoses of public health importance. Reports indicate human exposure to Rickettsial pathogens in Namibia through serological surveys, but there is a lack of data on infection rates in tick vectors, hindering the assessment of the relative risk to humans. Our study sought to screen Ixodid ticks collected from livestock for the presence of *Rickettsia* species in order to determine infection rates in ticks and to determine the *Rickettsia* species circulating in the country. We collected and pooled *Hyalomma* and *Rhipicephalus* ticks from two adjacent regions of Namibia (Khomas and Otjozondjupa) and observed an overall minimum *Rickettsia* infection rate of 8.6% (26/304), with an estimated overall pooled prevalence of 9.94% (95% CI: 6.5–14.3). There were no statistically significant differences in the estimated pooled prevalence between the two regions or tick genera. Based on the nucleotide sequence similarity and phylogenetic analysis of the outer membrane protein A (n = 9) and citrate synthase (n = 12) genes, BLAST analysis revealed similarity between *Rickettsia africae* (n = 2) and *Rickettsia aeschlimannii* (n = 11), with sequence identities ranging from 98.46 to 100%. Our initial study in Namibia indicates that both zoonotic *R. africae* and *R. aeschlimannii* are in circulation in the country, with *R. aeschlimannii* being the predominant species.

## 1. Introduction

Ticks are considered very important vectors of a variety of pathogens, including viruses, bacteria, and protozoa, which are of significant importance to animal and human health. Ticks have the ability to survive on multiple hosts in their lives and have a relatively long life span, which helps them to adapt and survive in different habitats [1]. Climatic and anthropogenic activities have resulted in a changed landscape of tick distribution geographically, with the creation of new ecological niches ideal for tick survival [1,2]. Climatic factors have also been shown to have a direct influence on the risk of pathogen transmission by ticks through their effect on pathogen survival in the tick and, more importantly, through the influence on tick questing behavior, which has a direct effect on the risk of pathogen transmission [3,4]. Consequently, a number of tick-transmitted pathogens are currently considered either emergent or re-emergent, due in part to the aforementioned change in tick distribution as well as an improvement in pathogen detection methods [2,5,6]. Among the tick-borne pathogens, the relative importance of *Rickettsia* species as an important pathogen, globally, has grown significantly in recent decades [1].

*Rickettsia*, which are considered endosymbiotic organisms of various arthropod species [7,8], can be broadly divided into five (5) basic groups, namely a basal ancestral group, the typhus group, the transitional group, the spotted fever group, and the Tamurae/Ixodes group [9]. With the advent and increasing use of molecular tools, the genus *Rickettsia* shows an increase in the number of newly reported species [10], with speciation based on the genetic divergence percentage in a number of genes [11]. The spotted fever group Rickettsiae (SFGR) are responsible for most human infections and is considered one of the most common emerging and re-emerging vector-borne pathogens globally [12,13]. Global interest in *Rickettsia* species has been shown by the increase in reports of new species in the last 30 years in different regions of the world [14], especially with new reports on human infections [15]. Of the reported SFGR, 21 of these are considered to be human pathogenic vector-borne pathogens [3,12,16].

SFGR are mostly transmitted by ticks, with the exception of *R. felis* and *R. akari*, which are transmitted by fleas and mites, respectively [17]. In ticks, transmission is known to occur transstadially [18] and/or transovarially [19], which makes ticks serve as vectors, reservoirs, and amplifiers of infections [13,20]. Therefore, the distribution and relative abundance of *Rickettsia* species in any geographical area are influenced by factors that affect tick distribution. A total of 146 tick species have been reported globally to carry SFGR, and about 13 SFGR species have been reported across the African continent [21]. Within the SFGR, *R. africae* and *R. aeschlimannii* exhibit a broad range of ticks that serve as vectors, hence their wide geographical distribution [21]. Even though *Amblyomma* species are considered the principal vectors of *R. africae* in sub-Saharan Africa [22], the bacterium has been reported to have varying infection rates in *Hyalomma* and *Rhipicephalus* tick species [23,24].

Despite reports of human infective *Rickettsia* species in ticks within the southern African region [24,25,26,27], and serological reports of human infections caused by Rickettsial pathogens in Namibia [28], there have been no molecular reports of *Rickettsia* circulation in the country. A previous study [29] conducted on small mammals in two regions of the country did not report the presence of *Rickettsia* species in this group of animals. Thus, the aim of this study was to screen for *Rickettsia* species in ticks from two neighboring regions of the country and characterize them to identify the species circulating in the area.

## 2. Materials and Methods

### 2.1. Study Sites

The study was conducted in the farming communities of the Khomas (Neudamm) and Otjozondjupa (Ovitoto) regions of Namibia (Figure 1). The two regions have semi-arid climatic conditions, with dry grassy plains. The Khomas region has more commercial farms, whereas the Otjozondjupa region is a resource-limited area with subsistence farming. Ticks were collected from cattle in the two sampling areas during the period of January to April 2023, and this was purposively carried out as they presented better chances of collecting significant numbers of ticks compared to environmental sampling. For the Khomas region, sampling was conducted on one farm (Neudamm) from animals that belonged to four (4) separate grazing groups. In the Otjozondjupa region, sampling was conducted in a village (Ovitoto) from a number of different herds that shared a common grazing area.

Ticks were collected from animals restrained in a crush pen using a pair of forceps and placed in aerated and humidified tubes for transportation to the laboratory. In the laboratory, the ticks were individually identified using established identification keys [30], and then placed in individual Eppendorf tubes and stored at −80 °C until further analysis. DNA was extracted from individual ticks using the Quick-DNA Miniprep Kit (Zymo Research, Orange, CA, USA), according to the manufacturer’s protocol, with the addition of a homogenization process for ticks that used a microbead beater. After the extraction process, DNA was pooled (4 ticks/pool) based on the genus of the tick and the sampling area.

### 2.2. Molecular Screening and Identification of Rickettsia Species

For initial screening for the presence of Rickettsia species, OneTaq^®^ Quick-load 2X Mastermix with standard buffer (New England BioLabs^®^ Inc., Ipswich, MA, USA) was used to amplify a 507-bp fragment of the outer membrane protein B (*omp*B) using primers designed for this study (Table 1). A known Rickettsia-positive sample (LC565644) [25] was included as a positive control, whereas purified water was included as a negative control. The PCR amplification reaction involved a one-step enzyme activation at 95 °C for 3 min, followed by 35 cycles of denaturation at 95 °C for 15 s, annealing at 46 °C for 30 s, extension at 72 °C for 45 s, with a final extension at 72 °C for 5 min. The amplified products were then electrophoresed on a 1% agarose gel, stained with ethidium bromide, and subsequently viewed under UV light.

All pools positive on the initial screening were subsequently selected for amplification of near-full-length fragments of the outer membrane protein A (*omp*A) (around 2971 bp) and citrate synthase (*glt*A) genes (around 1076 bp) using LongAmp^®^ Taq 2X Mater mix (New England BioLabs^®^ Inc., USA). The PCR amplification reaction involved a one-step enzyme activation at 94 °C for 30 s, followed by 30 cycles of denaturation at 94 °C for 15 s, annealing at 45 °C for 30 s, and extension at 65 °C for 6 min, with a final extension at 65 °C for 10 min. The amplified products were then electrophoresed on a 1% agarose gel, stained with ethidium bromide, and subsequently viewed under UV light to confirm the success of PCR amplification.

Near-full-length PCR products of the *omp*A and *glt*A genes were purified using the DNA Clean & Concentrator^®^ kit (Zymo Research) in preparation for sequencing using the Illumina platform (iSeq 100, San Diego, CA, USA). The libraries were prepared and sequenced following the Illumina Baym Library preparation protocol with modifications to the first-stage PCRs and the first PCR clean-up. The first-stage PCR primers were modified to include Illumina iSeq 100 adapter sequences. The Rickettsia-specific gene libraries were subsequently sequenced by paired-end sequencing using the iSeq™ 100 i1 Reagent v2 kit (300-cycle) chemistry (San Diego, CA, USA), according to the manufacturer’s instructions.

### 2.3. Phylogenetic Analysis

The raw Illumina sequence data were first subjected to quality control using FastQC (https://github.com/s-andrews/FastQC/, accessed on 7 January 2024), followed by trimming and barcode removal using SICKLE (https://github.com/najoshi/sickle, accessed on 10 January 2024). Following these pre-processing steps, de novo assembly was performed using SPAdes [31] (https://github.com/ablab/spades, accessed on 12 January 2024). The sequences obtained in this study were deposited in GenBank under accession numbers LC799204–LC799224. Near-full-length gene sequences of *glt*A (1076 bp) and *omp*A (2971 bp) were successfully obtained and subjected to standard nucleotide BLAST analysis on the NCBI website (http://www.ncbi.nlm.nih.gov/BLAST, accessed on 6 February 2024). Further, reference sequences were downloaded from GenBank, and together with the obtained nucleotide sequences, a multiple sequence alignment file was generated using Clustal W1.6 (GENETYX Corporation, Tokyo, Japan). Subsequently, the multiple sequence alignment file in fasta format was converted to a mega file and then employed to construct maximum likelihood phylogenetic trees. For *glt*A, the Tamura-3 parameter + Invariable sites model was used, whereas for *omp*A, the Tamura-3 parameter + Gamma distribution model was applied. Before constructing the phylogenetic trees, model selection for each gene was conducted using the model selection function available in MEGA 6. The phylogenetic tree construction was performed with a confidence level of 1000 bootstrap replicates [32] using the computer software MEGA XI [33].

### 2.4. Statistical Analysis

The results from the initial screening based on the *omp*B gene were used to determine the minimum infection rate (MIR), which was calculated as the proportion of tick pools that showed amplification of the target gene out of the total number of ticks tested multiplied by 100 [34]. Pooled prevalence estimates for perfect tests with exact confidence limits were calculated using EpiTools epidemiological calculators [35], assuming 100% test sensitivity and specificity for a fixed pool size (http://epitools.ausvet.com.au accessed on 12 December 2023). EpiTools was additionally used to compare the different estimated pooled prevalence based on the sampling site and genera of ticks.

## 3. Results

### 3.1. Tick Sampling and Identification

A total of 304 ticks were collected from the two (2) sampling areas (Neudamm—204 ticks; and Ovitoto—100 ticks). After identification up to the genus level using morphological keys [30], the ticks were subsequently pooled into a total of 76 pools based on the sampling area and genus. There were 51 pools from Neudamm and 25 pools from Ovitoto, of which 70 were of *Hyalomma* species and the remaining six (6) were of *Rhipicephalus* species.

### 3.2. Rickettsia Screening and Identification

During the initial screening for *Rickettsia*, an overall MIR of 8.6% (26/304) was observed. When segregated by sampling area, the MIR were 8.82% (18/204) and 8% (8/100) for Neudamm and Ovitoto, respectively. When segregated by tick genus, the MIR for *Hyalomma* ticks was 8.57% (24/280), whereas that for *Rhipicephalus* was 8.33% (2/24). On statistical analysis, the overall pooled prevalence was determined as 9.94% (95% CI: 6.5–14.3), with the regional pooled prevalences of 10.31% (95% CI: 6.15–15.88) and 9.91% (95% CI: 3.97–17.42) for Neudamm and Ovitoto, respectively. When segregated by genera, the estimated pooled prevalence for *Hyalomma* ticks was 9.96% (95% CI: 6.43–14.52), whereas that for the *Rhipicephalus* genus was 9.64% (95% CI: 1.1–31.3). The differences in estimated prevalence when compared across sampling sites and tick genera were not statistically significant (*p* = 0.91 and *p* = 0.96, respectively).

A random selection of 14 *Rickettsia*-positive pools was chosen for amplification and sequencing using the near-full-length genes of the *omp*A and *glt*A. Twelve of the samples were successfully sequenced based on the *glt*A gene (1054–1647 bp), whereas nine (9) of the samples were successfully sequenced based on the *omp*A gene (2912 bp). Standard nucleotide BLAST analysis of the obtained *glt*A sequences showed high similarity to *R. aeaschlimannii* (91.7%; 11/12) and *R. africae* (8.3%; 1/12), with sequence identity ranging from 99 to 100% (Table 2). Based on the *omp*A sequence BLAST analysis, *R. aeschlimannii* constituted 77.8% (7/9), with the rest (22.2%; 2/9) being *R. africae* with sequence identity ranging from 98.5 to 99.9%.

Further, phylogenetic analysis of the *glt*A gene (1078 bp) (Figure 2) and *omp*A gene (2912 bp) (Figure 3) revealed a close clustering of sequences based on species. Eleven sequences from Namibia closely clustered with *R. aeschlimannii* (accession number HQ335153), whereas one (1) exhibited close clustering with *R. africae* (accession number KX 819298) when considering the *glt*A gene (Figure 2). Similarly, considering the *omp*A gene, seven (7) and two (2) Namibian sequences closely clustered with *R. aeschlimannii* (accession number U83446) and *R. africae* (accession number U83436), respectively. There was agreement in species identity between the *glt*A and *omp*A sequences in all the samples (n = 8) that were successfully sequenced for both genes. Four (4) samples were successfully sequenced using only the *glt*A gene and not the *omp*A gene.

## 4. Discussion

In this study, we sought to investigate the presence of *Rickettsia* species in Ixodid ticks collected from cattle in two regions of Namibia. To the best of our knowledge, this is the first study to screen for *Rickettsia* species in ticks in Namibia, despite previous evidence of human exposure to this bacterium [28]. The only other study that sought to screen for *Rickettsia* species in Namibia was based on screening small mammals, and no molecular evidence of the pathogen was reported in that study [29]. The lack of previous studies on *Rickettsia* in ticks in Namibia is in contrast to the global trend, where studies on *Rickettsia* predominantly focus on ticks rather than on animals or humans [21]. Within sub-Saharan Africa, Rickettsia species have been reported in *Amblyomma*, *Hyalomma*, and *Rhipicephalus* tick species, with *Amblyomma* and *Hyalomma* considered the principal vectors [13]. In our study, the overall MIR was 8.6%, which is within the prevalence range (3–77%) reported in other studies on ticks conducted in countries bordering Namibia [25,26,27,36,37,38,39,40], with the variability hypothesized to be due to factors such as ecological differences (biotic and abiotic) in different locations [41]. The detection of *Rickettsia* species in ticks from our study, coupled with the serological evidence of exposure to SFGR in humans [28] and molecular reports of the pathogen in hyenas [42], indicates the existence of transmission cycles of *Rickettsia* species in the country.

While *Amblyomma* and *Hyalomma* tick species are considered the main vectors of *Rickettsia* species, with relatively higher infection rates than those observed in *Rhipicephalus* [13], in our study, the estimated pooled prevalence between *Hyalomma* and *Rhipicephalus* species was comparable and not statistically significant. Infections in *Rhipicephalus* species are generally considered to be concomitant or through co-feeding and/or feeding on a bacterial host [36,43]. Considering that the *Rhipicephalus* ticks sampled in our study were collected from cattle with co-infestations with *Hyalomma* ticks, this could explain the similarity in infection rates between the two tick species. However, it should also be noted that there are some reports that highlight the role of *Rhipicephalus* tick species as efficient vectors of *Rickettsia* species, with evidence of transovarial transmission also occurring in this tick genus [44,45]. This shows that there is still much more to learn about the vector–pathogen role and interactions in order to fully elucidate the transmission dynamics of tick-borne Rickettsial pathogens.

Within the southern African region, various tick species have been shown to harbor *Rickettsia* species at varying prevalence levels, and these *include Hyalomma marginatum rufipes*, *Hyalomma trancatum*, *Rhipicephalus appendiculatus*, *Rhipicephalus decoloratus*, and *Rhipicephalus evertsi evertsi* [24,27,37,46,47]. Our current study, however, did not specify the ticks sampled; thus, we recommend that future studies in the country show the prevalence of different tick species, as this will allow for a better understanding of the epidemiology of *Rickettsia* species in the region. This is especially important, as genetic differences in vectors are known to influence their vector competence [48]; thus, the relative importance of each tick species in the epidemiology of any specified pathogen.

There were no statistically significant differences between the infection rates of ticks collected from the two regions. This could be explained by the fact that the ecological settings of the two regions are similar, both being dry to arid regions, with a predominance of cattle being the main hosts for ticks. Differences in land use, climate, and environmental conditions are commonly considered factors that influence the prevalence of tick-borne pathogens [49,50], and these factors being similar in the two study sites could explain the observed similarity.

The speciation of *Rickettsia* species is generally based on multilocus sequencing and the agreement of a number of genes [11] or sequencing of long fragments of genes [51]. In our study, we amplified and sequenced near-full-length fragments of the *omp*A and *glt*A genes for speciation, thus giving us confidence in our sequence identities for those samples on which both genes were sequenced and had agreement on sequence identity for both genes. Based on the agreement in sequence identity on BLAST for the *omp*A and *glt*A genes, we report the presence of *R. aeschlimannii* and *R. africae* in Ixodid ticks in Namibia. Our study reports, for the first time, the molecular evidence of *R. africae* and *R. aeschlimannii* in Namibia, further showing the expanded geographical range of this bacteria. There was a significantly higher proportion of *R. aeschlimannii* in our study in comparison to *R. africae*, which could be due to the predominance of *Hyalomma* tick species, which are generally considered to be the principal vector for *R. aeschlimannii* [13].

In our study, there was a failure of amplification, and thus, there was no subsequent sequence analysis of the four (4) samples based on the *omp*A gene, despite these samples being successfully amplified and sequenced based on the *glt*A gene. It is thus possible that these samples which could not be sequenced on the *omp*A gene belong to a different genotype that could not be detected through this gene, as has previously been reported [52]. As such, we could not conclusively speciate these *Rickettsia* species, as we only had one sequence, thus not meeting the threshold for multilocus sequencing required for Rickettsia species [11]. It is recommended that future studies explore more genes than the two used in our study in order to allow for the speciation of all positive samples.

Rickettsial infections are common causes of acute febrile illnesses in travelers to sub-Saharan African regions [53]. Evidence of human infection by *Rickettsia* species has been reported in Namibia since 1986 [54], with serological evidence in the local population. Subsequent reports of human infections in the country were based on reports from international travelers [22,55]. In none of the reported human cases was the *Rickettsia* species responsible for the infection reported, leaving a gap in knowledge regarding the infective species circulating in the country. Both *Rickettsia* species reported in our study are human infective; therefore, the results of this study highlight the risk of human infection by these rickettsial pathogens in Namibia. There exists a possibility that the previously reported human cases could have been the result of either of the pathogens reported in our study. Reports of human infection with *R. africae* date as far back as the 1930s [56], even though the naming of the infectious agent was only confirmed in 1996 [57]. *R. aeschlimannii* is a more recently reported zoonotic pathogen, which was first reported in a traveler from Morocco [58], and since then, there have been several reports of human infections [59,60,61], further confirming the zoonotic potential of this bacteria. Considering the apparent circulation of these pathogens in farming communities, as well as the fact that infections result in non-specific febrile illnesses, it is important to consider *Rickettsia* species as a possible causal agent in patients showing such clinical signs.

## 5. Conclusions

We report the detection of *R. africae* and *R. aeschlimannii* in Ixodid ticks from two regions of Namibia. Our study is the first in the country to conduct a molecular survey of *Rickettsia* species in ticks and adds to the growing body of knowledge on the epidemiology of *Rickettsia* species in the region. The *Rickettsia* species identified in our study are both known to be zoonotic, which indicates a threat of infection to humans from these bacteria. Thus, there is a need for more epidemiological surveys to fully understand the epidemiology of these tick-borne bacteria in the country, including aspects such as determining if there is a clustering of pathogens by the host animal and/or sampling farms, surveys to define the relative importance of each tick species in the transmission of the pathogens, as well as to determine if there are other zoonotic *Rickettsia* species in circulation. There is considerable habitat sharing between humans and ticks, a factor that makes it very challenging to avoid contact with ticks. This highlights the importance of awareness campaigns on the public health significance of *Rickettsia* and tick-borne pathogens. Knowledge and understanding of the key human behaviors that increase the risk of contact with ticks and potential exposure to *Rickettsia* species is necessary at a localized level in order to design control programs for communities.

## Figures and Tables

**Figure 1 microorganisms-12-00912-f001:**
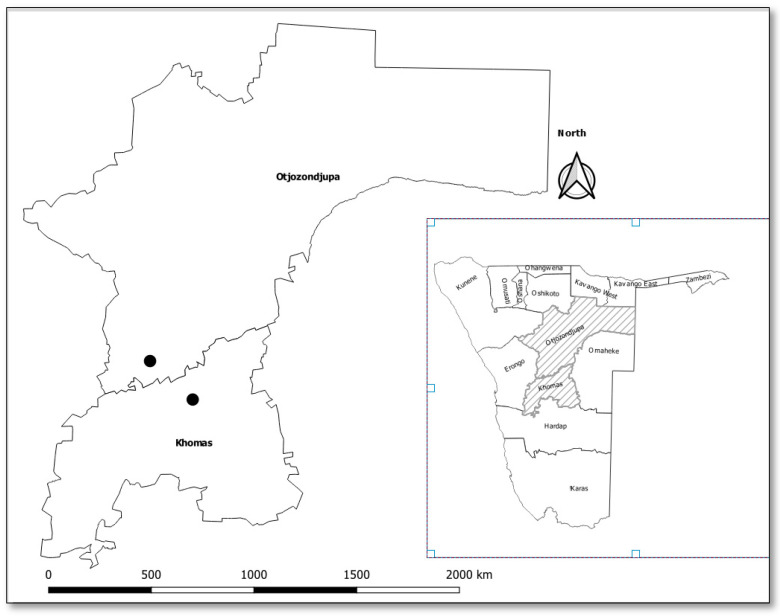
Map of Namibia showing the study sites; Ovitoto village in Otjozondjupa region and Neudamm farm in Khomas region.

**Figure 2 microorganisms-12-00912-f002:**
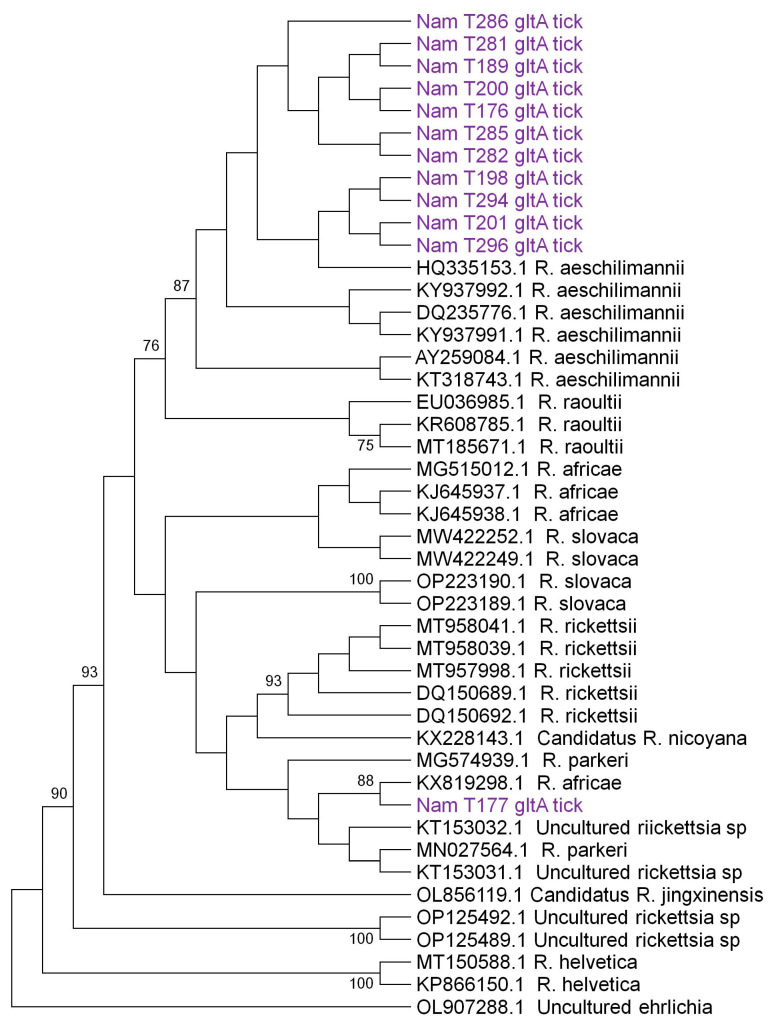
Maximum likelihood phylogenetic tree of *Rickettsia* spp. based on 1070 bp nucleotide sequences of the *glt*A gene. The tree was constructed using the Tamura-3 parameter + invariable sites model with 1000 bootstrap replicates as a confidence interval using MEGA XI. Bootstrap values less than 75% are not shown. Sequences from Namibia collected from ticks are highlighted in purple and have the prefix “Nam”.

**Figure 3 microorganisms-12-00912-f003:**
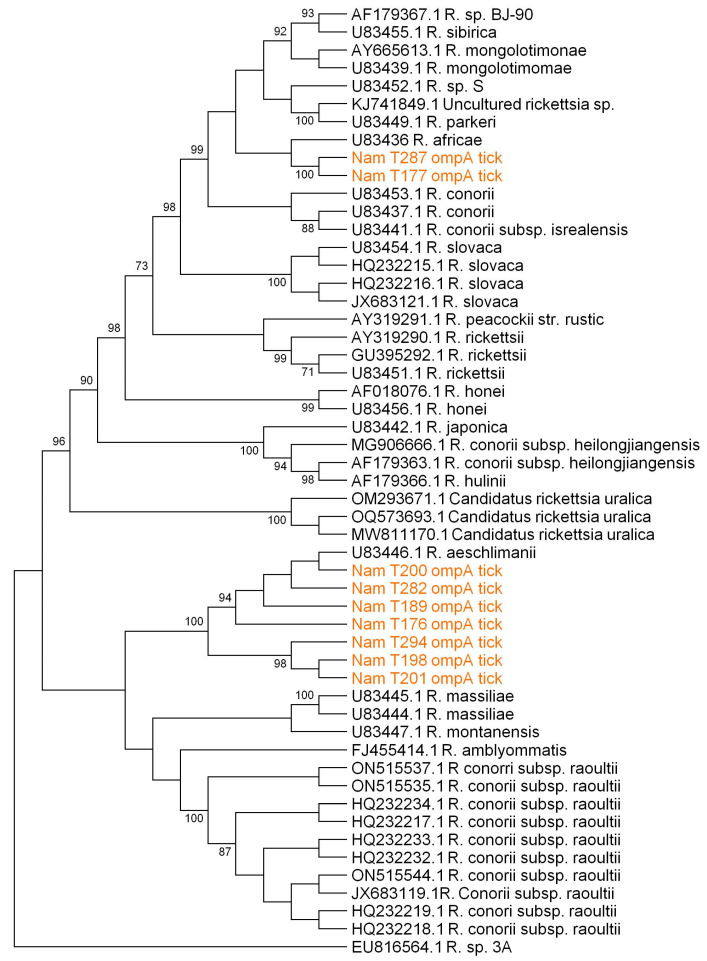
Phylogenetic tree for *Rickettsia* spp. generated using maximum likelihood analysis, utilizing 3170 base pair nucleotide sequences from the *omp*A gene. The Tamura-3 parameter + invariable sites model was employed for tree construction, incorporating 1000 bootstrap replicates as a confidence interval within MEGA XI. Bootstrap values below 75% are not displayed. Sequences from ticks collected in Namibia are highlighted in brown and have the prefix “Nam” in the tree.

**Table 1 microorganisms-12-00912-t001:** List of primer names and sequences used in this study.

Primer Name	Primer Sequence	Target Gene
Rick_ompB_2	GGTGTAGGAACAATAGACTT	*omp*B ^1^
Rick_ompB_R2	ATCTACGCTAACAACAAAT	
Rick_gltA_F1	TGCGGAAGCCGATTGCTTTA	*glt*A ^2^
Rick_gltA_R3	ATCCAGCCTACGGTTCTTGC	
Rick_ompA_F1	ACGGACCTCTTGATGGTGGT	*omp*A ^3^
Rick_ompA_R6	CCATTGCGTAAAGCTCAGGTG	

^1^ Outer membrane protein B; ^2^ citrate synthase; ^3^ outer membrane protein A.

**Table 2 microorganisms-12-00912-t002:** Table of sequence identity for samples sequenced in this study.

Sample ID	Outer Membrane Protein A (*omp*A) Gene				Citrate Synthase (*glt*A) Gene			
	BLAST Result	Accession Number	% Identity	Fragment Length (bp)	BLAST Result	Accession Number	% Identity	Fragment Length (bp)
T176	*R. aeschlimannii*	OR687032	99.9	2896	*R. aeschlimannii*	OR687023	99.91	1064
T177	*R. africae*	CP001612	98.92	2871	*R. africae*	HQ335126	99.91	1056
T189	*R. aeschlimannii*	OR687032	99.9	2900	*R. aeschlimannii*	OR687023	99.91	1054
T198	*R. aeschlimannii*	OR687032	99.55	2889	*R. aeschlimannii*	HQ335153	100	1057
T200	*R. aeschlimannii*	OR687032	99.72	2890	*R. aeschlimannii*	OR687023	99.9	1048
T201	*R. aeschlimannii*	OR687032	99.55	2891	*R. aeschlimannii*	HQ335153	100	1058
T281					*R. aeschlimannii*	OR687023	99.9	1277
T282	*R. aeschlimannii*	OR687032	99.93	2966	*R. aeschlimannii*	OR687023	99.91	1185
T285					*R. aeschlimannii*	OR687023	99.91	1245
T286					*R. aeschlimannii*	OR687023	99.91	1465
T287	*R. africae*	CP001612	99.24	2968				
T294	*R. aeschlimannii*	OR687032	99.62	3065	*R. aeschlimannii*	HQ335153	100	1647
T296					*R. aeschlimannii*	HQ335153	99.91	1073

## Data Availability

Data are contained in the article as the sequences are available on GenBank.

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
