# Peer review of "Molecular Detection and Characterization of Rickettsia Species in Ixodid Ticks from Selected Regions of Namibia"

_microorganisms, 2024, doi:10.3390/microorganisms12050912_

Round 1
Reviewer 1 Report
Comments and Suggestions for Authors
Line 99 “For Khomas region, sampling was done on one farm”; what was the criteria for selecting this farm? , do you consider that this farm is representative of the area?.
I suggest including a map where you can locate the sampled regions and know how far they are from each other.
Mention in the methodology what time of year the ticks were collected.
Line 146 Change the format of this reference (Bankevich et al., 2012).
Line 158 Change the format of this references (Tamura, 1992) and (Tamura et al., 2021).
In the references section it is suggested to consult the format requested by the journal and correct them.
Reviewer 2 Report
Comments and Suggestions for Authors
The main achievement of the article is that it is the first description of R. africae and R. aeschlimannii in Namibia, however the lack of information on the species of ticks carrying the agents diminishes the importance of the finding.
The methodology and results related to Rickettsia are correct, however the lack of data on the ticks from which they were obtained makes the work less interesting.
The team should have an entomologist for the prior identification of tick species. In Namibia, several species of the tick genera Hyalomma (H. rufipes, H. truncatum, H.turanicum) and Rhipicephalus (R. decoloratus, R. evertsi, R.simus, R.zambensi) have been described. Their morphological and physiological characteristics mean that there may be notable differences in their role as vectors of Rickettsiae. The specific identification of ticks is essential so that the information provided on the prevalence of infection has any meaning.
I encourage that although it can no longer be done morphologically, you analyze the tick species molecularly and redo the project with this information that will give relevance to the study.
Reviewer 3 Report
Comments and Suggestions for Authors
Dear authors,
Your manuscript “Molecular detection and characterization of Rickettsia species in Ixodid ticks from selected regions of Namibia” describes the detection and characterization of Rickettsia from Hyalomma and Rhipicephalus ticks collected from Khomas and Otjozondjupa of Namibia.
In current form, the manuscript could not be published and needs much revision.
First, English editing should be performed.
Second, I have some remarks:
The major remark - in order to draw a conclusion about the circulation of a pathogen, it is necessary, at a minimum. show its presence not only in vector ticks, but also in natural reservoirs. Therefore, your conclusion about the circulation of these two species of rickettsia is premature and not yet reliable.
Throughout the text, “Rickettsia” is sometimes written in italics, sometimes not...correct, please.
Why were these particular areas of Namibia chosen for study? Explanation needed.
Section 2.2 - “…with modifications to the first stage PCRs and the first PCR clean-up…” – please, provide more information about modification of the method.
L155-156 – “For gltA, the Tamura-3 parameter + Invariable sites model was used, whereas for ompA, the Tamura-3 parameter + Gamma distribution model was applied.” – How did you estimate the model appropriate for each gene sequences? The information should be added.
L187-194 - The approach you are using, namely, the use of at least two loci to identify the species of rickettsia under study is correct. However, it turned out that you identified 12 sequences at one locus, and only 9 at another. Could this mean that the sequences were not recognized by the primers you chose, and are actually a different rickettsia species than R. aeaschlimannii or R. africae? Or what is the cause of your sequencing failure?
Figure 1, 2 – based on appearance of the dendrograms, I am not sure that these dendrograms were built using ML method. Could you provide the tree file from MEGA, please?
Comments on the Quality of English LanguageEnglish editing should be performed.
Round 2
Reviewer 2 Report
Comments and Suggestions for Authors
Although actual corrections have improved the quality of the paper, the main issue has not been resolved, the lack of information on the ticks in which they were detected. We only know 76 pools of 4 ticks/pool where used: 51 of Nudamann and 25 of Ovitototo; 70 of them were Hyalomma ticks and 6 of Rhipicephalus, without specifying their geographic location. I understand that the genetic material is available and could be used for specific identification of the specimens. I suggest, as I did in my previous review than an entomologist should be included because possibly there are several species of each of the genera of ticks and some interesting information could be obtained.
Reviewer 3 Report
Comments and Suggestions for Authors
Dear authors,
Thank you for your response and revision of the manuscript. After that, I have no remarks.
Author Response
Dear Reviewer,
We thank you for your comments which allowed improvement of the manuscript. We are happy that we adequately addressed those comments.